# State of the art modelling for the Black Sea ecosystem to support European policies

**Natalia Serpetti**[1]*, **Chiara Piroddi**[1], **Ekin Akoglu**[2,3], **Elisa Garcia-Gorriz**[1], **Svetla Miladinova**[1], **Diego Macias**[1]

1 European Commission, Joint Research Centre, Directorate D-Sustainable Resources, Ispra, Italy, 2 Department of Marine Biology and Fisheries, Institute of Marine Sciences, Middle East Technical University, Erdemli-Mersin, Türkiye, 3 Climate Change and Sustainable Development Application and Research Center, Middle East Technical University, Ankara, Türkiye

* Natalia.Serpetti@ec.europa.eu

**Data Availability Statement:** All relevant data are within the paper and its Supporting information files.

## Abstract

The Black Sea is affected by numerous anthropogenic pressures, such as eutrophication and pollution through coastal and river discharges, fisheries overexploitation, species invasions, and the impacts of climate change. Growing concerns regarding the cumulative effects of these pressures have necessitated the need for an ecosystem approach to assessing the state of this basin. In recent years, the European Commission-JRC has developed a scientific and modelling tool, the Blue2 Modelling Framework with the aim of exploring the consequences of EU management and policy options on marine ecosystems. This framework has been designed to provide information on specific ecological indicators set out in EU legislation. Here, we present the Blue2 framework for the Black Sea ecosystem. The model represented the mid-1990s' conditions in the Black Sea ecosystem including trophic levels from primary producers to marine mammals and sea birds. The model simulations covered a period from 1995–2021. The results showed that gulls & cormorant seabirds, sprat, horse mackerel and mugilidae had structuring role in the food web. Fishing fleets had indirect negative impacts on marine mammals in addition to commercially exploited species. Analysis of the ecosystem indicators confirmed the overall temporal degradation of the Black Sea when comparing results with other Black Sea models, whilst the comparison with the Mediterranean Sea allowed us to identify comparable indicators between similar model structures. The spatial/temporal model successfully simulated the overall ongoing declining dynamics of the Black Sea ecosystem as the biomasses of the majority of the functional groups had significant observed decreasing trends during the simulation period. This model is the first attempt to represent the historical and current state of the Black Sea ecosystem spatially and temporally, serving as a reference baseline for evaluating policy scenarios and assisting policy makers in the evaluation of potential environmental impacts of management options.

## 1. Introduction

Since the 1960s, the Black Sea has undergone a series of transformations driven by numerous anthropogenic stressors. These stressors, acting either concurrently or sequentially, include

**Funding:** The author(s) received no specific funding for this work.

**Competing interests:** The authors have declared that no competing interests exist.

overfishing [1–5], pollution [6, 7], eutrophication and hypoxia [8, 9], coastal abrasion and erosion [10, 11], invasion of alien species [5, 12, 13], and climate change [14].

The food web dynamics in the Black Sea have undergone significant changes over decades. In the 1960s, the system was predominantly controlled by top-down predator regulation, with species such as Atlantic mackerel (*Scomber scombrus*, Linnaeus, 1758), bluefish (*Pomatomus saltatrix*, Linnaeus, 1766), and Atlantic bonito (*Sarda sarda*, Bloch, 1793) being dominant. However, by the 1970s, there was a notable shift towards bottom-up resource control [15], largely attributed to concurrent eutrophication and the overexploitation of apex predators [16]. Recent reports indicate that over 60% of fish stocks are subjected to overfishing in the Mediterranean and the Black Seas [2, 17].

The anthropogenic impacts have significantly diminished both benthic and pelagic biodiversity, while undermining ecosystem resilience. By the late 1980s, the ecosystem, once dominated by small pelagic fish such as anchovy (*Engraulis encrasicolus*, Linnaeus, 1758), underwent another transformation marked by the collapse of the anchovy stock and the proliferation of the alien comb jelly species (*Mnemiopsis leidyi*, Agassiz, 1865) [3, 13]. Within a decade from the early 1980s, fisheries yield plummeted by 70% [18].

In the 1990s, active measures were taken to address overfishing and eutrophication [19]; however, despite these interventions, the overall health of the Black Sea ecosystem continues to deteriorate [16]. Climate-induced forces threaten this weak equilibrium [20]. The Black Sea one of the most impacted large marine ecosystems due to anthropogenic pressures [16] and one of the least spatially-assessed marine regions in relation to its biodiversity [21, 22].

Temporal-spatial analysis of species distributions and biodiversity is lacking for the Black Sea, despite the necessity to implement/enforce management policies. Cooperation and commitment from all countries bordering the Black Sea is required to ensure the long-term health and resilience of this ecosystem. To date, the ecosystem-based management (EBM) approach in the Black Sea has been limited to major commercial fish stocks for fisheries management advice [23].

Policymakers often embrace ecological modelling approaches to assess the impacts of decision-making processes in marine ecosystems [24]. In an evidence-based policy context, policymakers urge spatially and temporally explicit marine ecosystem assessments to develop comparable indicators across regions impacted by multiple stressors cumulatively. In this context, the General Fisheries Commission for the Mediterranean (GFCM) continue to make a great effort in data collection (extensively used in our model set-up) involving more countries on data sharing. Moreover, the European Commission and its science Directorate-General (DG), the Joint Research Centre, has developed the Blue2 modelling framework (Blue2MF), which consists of a comprehensive suite of modelling tools designed to simulate EU marine ecosystems, covering all essential components, across various management and policy scenarios [25]. The Blue2MF is extensively used to support EU policies, mainly the Marine Strategy Framework Directive (MSFD) [26–28] and the Zero Pollution Action Plan [29], by analysing the probable future environmental conditions in EU marine regions under different policy scenarios.

The Ecopath with Ecosim food-web modelling [30, 31], which is part of the Blue2MF, has been considered one of the most widely adopted High Trophic Level (HTL) approach to assess cumulative effects, in time and space, of multiple environmental conditions accounting for food-web interactions, anthropogenic stressors such as, fisheries, aquaculture, renewable energy and infrastructures [32, 33] and climate/environmental changes [34].

In this study, we present the Blue2MF for the Black Sea ecosystem and provide an initial spatio-temporal assessment of the Black Sea ecosystem since 1995 to delineate the recent conditions of the food web and fish stocks in the Black Sea. The model was set up with a structure

to encompass all representative species of the ecosystem [16, 23], while also following the rationale used by Piroddi et al. [24] for the Blue2MF of the Mediterranean Sea ecosystem, i.e., incorporating functional groups and/or species that represent the overall biodiversity of the ecosystem, with the same temporal range (1995–2021) and spatial resolution. Having similar ecological, temporal and spatial structure, enables comparison among models, in this case the Black and Mediterranean seas, and indicators, which, occasionally, are sensitive/dependent of model structure [35, 36]. The following rationale is important to provide independent and evidence-based support in the whole policies cycle assessment.

## 2. Materials and methods

The Blu2MF Black Sea model developed here is a comprehensive suite of modelling tools focusing on different ecosystem components, including atmospheric forcing, freshwater hydrology, marine oceanography, biochemical components, and the marine food web of the ecosystem [25].

BlueMF integrates models for freshwater quantity, i.e., LISFLOOD [37], and quality, i.e., GREEN [38]) to replicate the conditions of EU rivers, along with atmospheric forcing derived from either reanalysis sources (such as the ERA-5 dataset) or circulation models (such as Euro-CORDEX), which account for the atmospheric deposition of critical chemical elements for marine ecosystems, e.g., nitrate from the EMEP model [39]. The core of Blue2MF is an ocean model that consists of a hydrodynamic and a biogeochemical module (also called the lower trophic level (LTL)). The hydrodynamic component is founded on the General Estuarine Transport Model (GETM) [40] and the General Ocean Turbulence Model, whereas the biogeochemical module has been expertly customized for each EU marine region [41]. At the end of this framework, there is the higher trophic level (HTL) food web model (Ecopath with Ecosim (EwE) [30], which is coupled (offline) to the hydrodynamic biogeochemical module of the Blue2MF (S1 Fig in S1 Appendix).

In the Mediterranean Sea, the LTL of Blue2MF has been widely utilized to simulate biogeochemical conditions during hindcast periods [42, 43] and forecast future conditions [44, 45]. The HTL component has been utilized on multiple occasions to investigate the HTL conditions in this basin [24, 26, 46].

Following this rationale, the marine Black Sea model of Blue2MF consisted of an HTL food web model coupled with its Black Sea LTL hydrodynamic biogeochemical model (GETM-BSEM [47]. In particular, biomass and growth rate data for small and large phytoplankton and zooplankton were used as input data for Ecopath (the static mass-balance module, representing the year 1995), Ecosim (the temporal dynamic module representing the period 1995–2021)) and Ecospace (the spatial dynamic module representing the period 1995–2021).

### 2.1. Ecopath

The model was built in Ecopath with Ecosim (EwE) version 6.7. Ecopath is a mass balance model across functional groups (FGs) that represents a snapshot of the ecosystem in a given year. FGs are represented by species or groups of species that share similar predator-prey interactions, temporal biomass trends, ecosystem functioning, and habitat preferences.

FGs were selected based on the composition of the Black Sea commercial fishery [20] and ecosystem dynamics, as observed by Akoglu [16]. In addition, to assess and compare ecosystem structures across MSFD regions, the model was constructed following the same rationale described by Piroddi et al. [24], aiming to represent the diversity of the overall ecosystem and support best policy decisions.

Thus, in contrast to previously built Black Sea models [16, 23] top-predator mammal species (bottlenose dolphin, common dolphin and harbour porpoise), seabird species (seagulls & cormorants, terns and pelicans), benthic invertebrates ('mobile' and 'sessile' benthos groups) as well as seagrass and seaweed primary producers were included as FGs.

Input data (biomass, production/biomass, consumption/biomass, diet, and catch/discards) were extracted and/or calculated from in situ literature, stock assessments, or, when lacking, inferred from the Mediterranean model [24].

Catches (1995–2021) from the General Fisheries Commission for the Mediterranean (GFCM) (https://www.fao.org/gfcm/data/capture-production/ar/) were extracted to obtain a list of species that mainly represented the commercial species in the ecosystem. The species were subsequently organized within the defined functional groups (FGs, Table 1). Catches data were separated into landings and discards, by functional groups across the fishing fleets (Beach-Seiner, Trawler, Dredge, Gillnet and Trammel net, Longline, Traps and pods, Purse Seine and Midwater Trawl) following the catch and discard proportions used by Piroddi et al. [24] for the Eastern Mediterranean region, and when available, specific local fishery data by fleets were included (details in the S1 Table in S1 Appendix).

The biomasses of Atlantic bonito and bluefish were extracted from the literature and scaled down to consider the migratory behaviours of these stocks as they move into the Marmara and the Mediterranean Seas for wintering [4]. This approach was also applied to the Atlantic mackerel group. To account for changes in biomass, either temporally, outside the period considered, (i.e., species overfished before the model selected year) or spatially, outside the model domains (i.e., migration), the model allows to include a biomass accumulation term, negative or positive to represent the "Non-Steady-State Condition" of those species [31]. We applied a positive biomass accumulations for those species with migratory behaviours (Atlantic bonito, Bluefish, Mackerels), whilst we set a negative biomass accumulations for rapa whelk (*Rapana venosa*, Valenciennes, 1846) and picked dogfish (*Squalus acanthias*, Linnaeus, 1758) to better represent the temporal decline observed by their stock assessments (S1 Table in S1 Appendix).

The biomasses for all the other fished groups in 1995 was inferred using catch per unit of effort data (catch from the GFCM and reconstructed effort (for details, see section below)). Catches per Unit of Effort (CPUE) measured only the component of the population that is vulnerable to the gear (gear selectivity, size and age of fish, horizontal and vertical distribution of fish); to account for the total population, we corrected it applying a coefficient factor estimated by the model. For benthic invertebrates, data from literature were used (S1 Table in S1 Appendix).

Ecopath is based on the assumption that the production of one FG is equal to the sum of all predation, non-predatory losses, exports, biomass accumulations, and catches, as expressed by the following equation:

$$P/B_i \cdot B_i = P/B_i \cdot B_i \cdot (1 - EE_i) + \sum_j (Q/B)_{ji} \cdot B_i \cdot DC_{ji} + Y_i + NM_i + BA_i \qquad (1)$$

where $B_i$ is the initial biomass (t*km$^{-2}$), $(P/B)_i$ is the production rate per unit biomass of group $i$ (year$^{-1}$), $(Q/B)_i$ is the consumption rate per unit biomass of group $i$ (year$^{-1}$), $DC_{ji}$ is the relative proportion of prey $i$ in the diet of predator $j$, $NM_i$ is the net migration of prey $i$, $BA_i$ is the biomass accumulation of prey $i$, $Y_i$ is the yield (landings + discards, t*km$^{-2}$year$^{-1}$) of prey $i$, and $EE_i$ is the ecotrophic efficiency of group $i$, that is, the proportion of production used in the system.

In an Ecopath model, the energy input and output of all the FGs must be balanced under certain ecological and thermodynamic rules. To balance the Black Sea model, we applied a

**Table 1. Model input parameters of the Black Sea balanced model for 1995.**

| Num | Group name | TL | B | P/B | Q/B | EE | P/Q | BA |
|---|---|---|---|---|---|---|---|---|
| 1 | Bottlenose dolphin | 4.23 | 0.013 | 0.07 | 12.45 | 0.151 | 0.01 | 0 |
| 2 | Common_dolphin | 4.29 | 0.016 | 0.04 | 13.65 | 0.252 | 0.00 | 0 |
| 3 | Harbour porpoise | 4.21 | 0.012 | 0.03 | 16.09 | 0.554 | 0.00 | 0 |
| 4 | Seagulls & Cormorants | 3.38 | 0.0009 | 0.34 | 192.63 | 0.017 | 0.00 | 0 |
| 5 | Terns | 3.36 | 0.0001 | 0.30 | 173.45 | 0.073 | 0.00 | 0 |
| 6 | Pelicans | 3.69 | 0.0002 | 0.42 | 54.32 | 0 | 0.01 | 0 |
| 7 | Other large pelagic fish | 3.80 | 0.005 | 0.77 | 3.71 | 0.192 | 0.21 | 0 |
| 8 | Atlantic bonito | 3.89 | 0.072 | 0.59 | 7.55 | 0.521 | 0.08 | 0.011 |
| 9 | Bluefish | 3.71 | 0.043 | 0.87 | 5.34 | 0.391 | 0.16 | 0.006 |
| 10 | Mackerels | 3.65 | 0.130 | 0.78 | 4.51 | 0.792 | 0.17 | 0.019 |
| 11 | Medium pelagics | 3.77 | 0.055 | 0.81 | 7.21 | 0.630 | 0.11 | 0 |
| 12 | Horse mackerel | 3.44 | 0.229 | 0.92 | 8.35 | 0.890 | 0.11 | 0 |
| 13 | Anchovy Ad. | 3.23 | 1.694 | 1.78 | 9.68 | 0.563 | 0.18 | 0 |
| 14 | Anchovy Juv. | 3.23 | 0.300 | 1.47 | 22.71 | 0.694 | 0.06 | 0 |
| 15 | Sprat | 3.21 | 0.611 | 1.08 | 11.02 | 0.846 | 0.10 | 0 |
| 16 | European pilchard | 3.11 | 0.067 | 1.39 | 9.47 | 0.828 | 0.15 | 0 |
| 17 | Other small pelagic fish | 3.29 | 0.029 | 1.14 | 5.96 | 0.707 | 0.19 | 0 |
| 18 | Large demersals | 3.79 | 0.011 | 0.42 | 2.51 | 0.501 | 0.17 | 0 |
| 19 | Medium demersal benthopelagic fish | 3.21 | 0.052 | 0.88 | 4.48 | 0.736 | 0.20 | 0 |
| 20 | Whiting | 3.53 | 0.080 | 1.13 | 5.15 | 0.880 | 0.22 | 0 |
| 21 | Mugilidae | 2.30 | 0.098 | 0.67 | 5.76 | 0.875 | 0.12 | 0 |
| 22 | Sparidae | 3.30 | 0.022 | 0.71 | 6.35 | 0.414 | 0.11 | 0 |
| 23 | Turbot | 2.97 | 0.011 | 0.71 | 4.66 | 0.761 | 0.15 | 0 |
| 24 | Flatfish | 2.98 | 0.015 | 0.61 | 3.92 | 0.973 | 0.15 | 0 |
| 25 | Red mullet | 3.18 | 0.053 | 0.99 | 6.62 | 0.777 | 0.15 | 0 |
| 26 | Striped mullet | 3.23 | 0.042 | 1.12 | 7.05 | 0.894 | 0.16 | 0 |
| 27 | Small demersal | 2.95 | 0.110 | 0.93 | 9.27 | 0.749 | 0.10 | 0 |
| 28 | Picked dogfish | 3.62 | 0.018 | 0.30 | 2.95 | 0.396 | 0.10 | -0.001 |
| 29 | Demersal sharks | 3.63 | 0.020 | 0.30 | 2.75 | 0.731 | 0.11 | 0 |
| 30 | Rays | 3.46 | 0.030 | 0.20 | 2.81 | 0.493 | 0.07 | 0 |
| 31 | Benthic cephalapods | 3.31 | 0.035 | 2.06 | 6.88 | 0.694 | 0.30 | 0 |
| 32 | Shrimp/prawns | 2.70 | 0.107 | 3.46 | 11.53 | 0.845 | 0.30 | 0 |
| 33 | Other decapods | 2.85 | 0.114 | 3.54 | 11.79 | 0.735 | 0.30 | 0 |
| 34 | Rapana venosa | 2.44 | 0.367 | 0.78 | 7.98 | 0.935 | 0.10 | -0.055 |
| 35 | Gastropoda | 2.00 | 0.541 | 1.82 | 15.00 | 0.494 | 0.12 | 0 |
| 36 | Bivalvia | 2.00 | 1.630 | 1.25 | 10.04 | 0.923 | 0.12 | 0 |
| 37 | Mobile benthos | 2.00 | 0.891 | 2.86 | 26.03 | 0.570 | 0.11 | 0 |
| 38 | Sessile benthos | 2.17 | 0.740 | 1.95 | 14.79 | 0.647 | 0.13 | 0 |
| 39 | Jellyfish | 2.91 | 1.565 | 3.94 | 27.92 | 0.024 | 0.14 | 0 |
| 40 | Zoo large | 2.26 | 6.525 | 70.59 | 220.59 | 0.756 | 0.32 | 0 |
| 41 | Zoo small | 2.00 | 0.715 | 52.15 | 155.65 | 0.521 | 0.34 | 0 |
| 42 | Phy large | 1.00 | 10.838 | 102.51 | | 0.805 | | 0 |
| 43 | Phy small | 1.00 | 2.405 | 156.39 | | 0.682 | | 0 |
| 44 | Seagrass | 1.00 | 0.696 | 2.53 | | 0.036 | | 0 |
| 45 | Seaweed | 1.00 | 1.091 | 1.53 | | 0.310 | | 0 |
| 46 | Discards | 1.00 | 0.074 | | | 0.894 | | 0 |

<div align="right">(<i>Continued</i>)</div>

**Table 1.** (Continued)

| Num | Group name | TL | B | P/B | Q/B | EE | P/Q | BA |
|---|---|---|---|---|---|---|---|---|
| 47 | Detritus | 1.00 | 22.928 | | | 0.208 | | 0 |

TL, trophic level; B = Biomass (t km$^{-2}$); P/B = Production/biomass (year$^{-1}$); Q/B = Consumption/biomass (year$^{-1}$); EE = Ecotrophic Efficiency, P/Q = Production/consumption (year$^{-1}$); BA = Biomass accumulation (t km$^{-2}$).

manual mass-balance procedure following Mackinson and Daskalov [48], which modifies the appropriate input parameters employing six criteria organised by trophic levels (values of ecotrophic efficiency, production/consumption rate, respiration/biomass rate, slope of the biomass, slope of the production/biomass rate and slope of the consumption/biomass rate) (S2 Fig in S1 Appendix). We balanced the model following the best practice guidelines [36, 49, 50]: mass balance was obtained primarily by adjusting the diet matrix and the coefficient factor used to estimate biomasses from the CPUE which are the data source with higher uncertainty. Here we presented selected factors, among all parameters, defining the ecological and thermodynamic rules for balancing Ecopath models.

**2.1.1. Ecosystem indicators.** The trophic flows, encompassing total production, consumption, respiration, catches, and flow to detritus, were estimated to represent the ecosystem structure and exploitation status [51–53]. The trophic level (TL) of each functional group was calculated as

$$TL_j = 1 + \sum_{i=1}^{n} DC_{ij} * TL_i \tag{2}$$

where '$j$' is the predator of prey '$i$', '$DC_{ji}$' is the fraction of prey '$i$' in the diet of predator '$j$', and $TL_i$ is the TL of prey i. By definition, TL I was attributed to primary producers and detritus, TL II to herbivores, TL III to first-order carnivores, and TL IV to second-order carnivores. This enabled us to calculate the TL of the catches ($TL_C$), which is an indicator of "fishing down the food web" effect in the ecosystems [54] as overfishing usually leads to a decrease in the overall trophic level of the catch. $TL_C$ was calculated as

$$TL_{Ci} = \frac{\sum_{i=1}^{n} TL_i * Y_i}{\sum_{i=1}^{n} Y_i} \tag{3}$$

where '$Y_i$' refers to the landings of species (group) '$i$'.

The total system throughput (TST) was calculated as the sum of all flows in the food web and is an indication of the whole ecosystem size. The total primary production/total system respiration (TPP/ TR) is a metric of system maturity and was calculated to assess the state of the ecosystem *sensu* [51]. The mean transfer efficiency (TE) is the efficiency at which energy is transferred between TLs and an indicator of the functioning of the food web, i.e., how efficiently the energy produced by lower trophic levels are transferred to higher trophic levels. The mean TE was calculated as the geometric mean of the TE for each integer TLs II to IV.

The system omnivory index (SOI) is based on the average omnivory index (OI), which is calculated as the variance of the trophic levels of a consumer's prey groups, indicating predatory specialization [53]. A high SOI is desired in mature ecosystems.

Relative system ascendency and overhead were also calculated. Ascendency is a measurement of system growth and the development of network links [55]. Overhead (O) is the energy in the reserve of an ecosystem that reflects the system's strength when it experiences unexpected perturbations [52].

Mixed trophic impact (MTI), which quantifies the impacts of a theoretical change in a specific group's biomass (including fishing activities) on other groups in the ecosystem [56], assessing both direct (e.g., predation) and indirect (e.g., competition) trophic relationships, was also calculated as

$$MTI_{ij} = DC_{ij} - FC_{ij} \tag{4}$$

where 'DC$_{ij}$' is the diet composition term expressing how much '$j$' contributes to the diet of '$i$', and 'FC$_{ji}$' is the proportion of predation on '$j$' that is due to '$i$' as a predator. MTI scales between -1, indicating a strong negative impact, and 1, indicating a strong positive impact, between FGs in the food web.

Finally, the keystoneness index (KS, Libralato et al. [57], which shows the FGs with a disproportiantely influential role in the food web contrary to their biomasses, was calculated as

$$KS_i = \log(\varepsilon_i * \frac{1}{p_i}) \tag{5}$$

where '$\varepsilon_i$' is the overall effect expressed as the square root of the sum of '$m_{ij}$' square (with '$m_{ij}$' being the relative impact of a slight increase in biomass of impacting group '$i$' on biomass of impacted group '$j$'), and $p_i$ is the contribution of the functional group to the total biomass of the food web.

## 2.2. Ecosim

Ecosim uses Ecopath's outputs as starting information to simulate the dynamics of each FG over time. In Ecosim, a series of differential equations describe the changes in biomass for each FG (i) over time, as follows:

$$\frac{dB_i}{dt} = \left(\frac{P}{Q}\right)_i \cdot \sum Q_{ji} - \sum Q_{ji} + I_i - (M_i + F_i - e_i) \cdot B_i \tag{6}$$

Where $dB_i/dt$ is the biomass change of group $i$ in the time step $dt$, $(P/Q)_i$ is its production/consumption ratio, i.e., gross growth efficiency, $Q_{ji}$ is the consumption of group $j$ (predator) on prey group(s) $i$, $Q_{ij}$ is the consumption for predation by all predators $j$ on group $i$ (prey), $I_i$ is the immigration rate, $B_i$ is the biomass at time $t$, $M_i$ and $F_i$ are the natural and fishing mortality rates of group $i$, respectively, and $e_i$ is the density-dependent emigration rate. Fishing mortality or fishing effort is used to drive the model.

Ecosim predicts biomasses and catches and compares them to observed time-series data of biomasses and catches using a log-likelihood sum of squares for calibration [58]. For each predator, consumption ($Q_{ij}$) is calculated based on the "foraging arena" theory, which divides the biomass of the prey into a vulnerable and a non-vulnerable fraction and the transfer rate or vulnerability between the two fractions determines the trophic flow between the predator and the prey [59]. The vulnerability concept incorporates density-dependent processes and expresses how far a group is from its carrying capacity. Default values of vulnerability (v = 2) represent a mixed trophic flow, a low value (v < 2) indicates a 'bottom-up', donor-driven, flow control and a situation closer to carrying capacity, while a high value (v > 2) indicates a 'top-down', predator-driven, flow control and a situation further away from carrying capacity. For each predator-prey interaction, the consumption rates are calculated as:

$$Q_{ij} = \frac{a_{ij} \cdot v_{ij} \cdot B_i \cdot B_j \cdot T_i \cdot T_j \cdot M_{ij}/D_j}{v_{ij} + v_{ij} \cdot T_i \cdot M_{ij} + a_{ij} \cdot M_{ij} \cdot B_j \cdot T_j/D_j} \cdot f\left(Env_{function}, t\right) \tag{7}$$

where $a_{ij}$ is the rate of effective search for prey *i* by predator *j*; $T_i$ represents prey's relative feeding time; $T_j$ is the predator's relative feeding time; $M_{ij}$ is the mediation forcing effects due to the presence of a third indirectly interacting species with the prey-predator pair in the food web; $v_{ij}$ is the vulnerability parameter, which is akin to the half-saturation constant in Holling-type functional responses [60]; $D_j$ represents the effects of handling time as a limit to consumption rate; and *f(Envfunction,t)* is the environmental response function that restricts the size of the foraging arena (or consumption) to account for external environmental drivers changing over time [34, 61]. Species-specific response functions to environmental factors were defined by minimum, 10th and 90th percentiles and maximum values of their optimal niches extracted from AquaMaps (https://www.aquamaps.org/) within the Black Sea domain (S3 Table in S1 Appendix). For functional groups defined by multiple species, the parameters were calculated as weighted means by species biomass composition.

For the Black Sea model, we used relative temporal changes in fishing efforts by fleet and net primary productivity for small and large phytoplankton, derived from the Blue2MF hydro-dynamic biogeochemical model (GETM-BSEM [47]) to drive the model and predict temporal biomass and catch changes (1995–2021).

"Fishing effort" (kW*days$^{-1}$) was calculated as the product of the number of fishing vessels, Kilowatt (kW) per vessel, and the number of days spent fishing. For EU countries, we used data from the official European Commission databases (STECF https://stecf.ec.europa.eu/index_en), while for other countries, we considered the available literature [2, 62–66]. To account for improvements in technology (e.g., mobile phones, GPS, sonar, radio) that were not captured by kW as a measure of effort, a conservative technological "creep factor" of approximately 0.8% per year was used, as observed by Damalas et al. [67] and the European Court of Auditors [68].

Stock assessment data were used to validate biomasses and catches for turbot, whiting, red mullet, picked dogfish, horse mackerel, sprat, anchovy, and rapa whelk, while the International Commission for the Conservation of Atlantic Tunas (ICCAT) database (https://www.iccat.int/en/accesingdb.HTML) was used for large pelagic fish. The model was fitted [69] versus time series of observations of relative biomasses for stock-assessed species and relative catches for all fished functional groups (reported by the GFCM database). The fitting procedure is a routine that minimizes the sum of squares (SS) ratio between predicted and observed data as a metric for assessing the model performance. The SS per functional group can be used to evaluate and compare the quality of the group's fitting to its respective reference time series data (for equal weight and number of observation points across the time-series). The fishing effort, primary production (PP) of large and small phytoplankton groups over time were used as the main forcing time series to drive the model.

The fitting procedure followed the approach described by Mackinson [58]. This method uses the Akaike Information criterion (AIC) [70, 71]:

$$AIC = nlog\,(minSS/n) + 2k \qquad (8)$$

where *n* is the number of observations, *minSS* is the minimum sum of squares resulting from the comparison of predicted with observed datasets, and *k* is the number of parameters to test statistical hypotheses related to changes in predator-prey dynamics, changes in productivity, impact of fishing, and possible combinations of the above-mentioned factors. The AIC is a statistical measure for model selection and penalizes for fitting too many parameters. AIC is used to choose the "best" model (the one yielding the lowest AIC) considering a good fit and the least number of estimated parameters. In this study, we used the second-order Akaike

Information Criterion (AIC$_c$) calculated as follows:

$$AICc = AIC + 2k(k-1)/(n-k-1) \tag{9}$$

to account for small sample sizes (n = observations) in the dataset.

Thirty-nine independent time series were used during the calibration process using different weights assigned to the time series to reflect the uncertainties of the data. We identified the best-fitted model using "by-predator" setting; additional vulnerabilities were found using "by-predators-preys" setting with a hybrid fitting approach [72, 73]. A thousand Monte Carlo runs were performed with the "best" fitted model to assess the uncertainties (5th and 95th percentiles) of the predictions using 10% variability in the input data (B, P/B, Q/B, EE).

Statistical significance of the changes in the temporal trends of simulated biomasses were analysed with time series and trend analyses. First, time series of simulated biomasses for the functional groups with observed data were decomposed into their seasonal, trend and irregular components using moving averages. The time series were then detrended by subtracting the time series of seasonal component from the time series of biomasses using the "stats" package in R [74]. Finally, nonlinear trend analysis was done using the "zyp" package in R [75] and the data were pre-whitened prior to trend analysis to eliminate autocorrelation as model simulated values are known to be hampered from autocorrelation following the Zhang's method [76].

## 2.3. Ecospace

Ecospace is the spatiotemporal dynamic module of EwE and, as for Ecopath and Ecosim modules, it is coupled with the hydrodynamic biogeochemical model GETM-BSEM outputs, namely depth, annual bottom oxygen concentration, temperature, salinity and monthly primary/secondary productivity (small and large phytoplankton and zooplankton) which drive the spatial distributions.

Since Ecospace does not explicitly account for dynamics in the water column [77], we extracted data on salinity, temperature, primary productivity and planktonic biomass for distinct depth ranges: surface (0–10 m), euphotic zone (0–50 m), entire water column (0–2000 m), and (4) bottom (maximum depth). All species are distributed in space according to their dispersal rates (S2 Table in S1 Appendix) in a two-dimensional spatial grid (~3.7 km x 3.7 km) in relation to their depth-optimal niches (S3 Table in S1 Appendix) from the coastline and to a maximum depth of 2212 m (Fig 1A). The bottom oxygen concentration was used to drive the benthic species distributions to avoid their distributions from anoxic bottom conditions (Fig 1B) (< 10 mmol O$_2$/m$^2$). Salinity integrated over 50m depth was only applied to the bivalve group, as values below 10‰ have a negative influence on the Mediterranean mussel, *Mytilus galloprovincialis* [78], the dominant species within this functional group. For the model hindcast, temperature and salinity (Fig 1C and 1D) were not used, with the above exception. S3 Table in S1 Appendix shows the response functions to environmental variables; in bold, the functions applied in the hindcast.

Spatiotemporal changes in planktonic groups (small and large phytoplankton and zooplankton) were used in Ecospace as absolute biomass (Fig 1E–1H), whereas the Net Primary Production of 1995 was used in the Ecospace base maps (Fig 1I).

Ecospace simulations were run with 5 years spin-up, a burn-in period with no forcing or disturbances that allows species to be distributed in space in relation to their environmental niches before applying the temporal changes in planktonic biomasses and fishing effort [77].

Within the Ecospace module, the Habitat Foraging Capacity (HFC) allows to assess the suitability areas of the modelled species/functional groups based on the cumulative effects of the applied environmental conditions [79]. The HFC maps were extracted and qualitatively

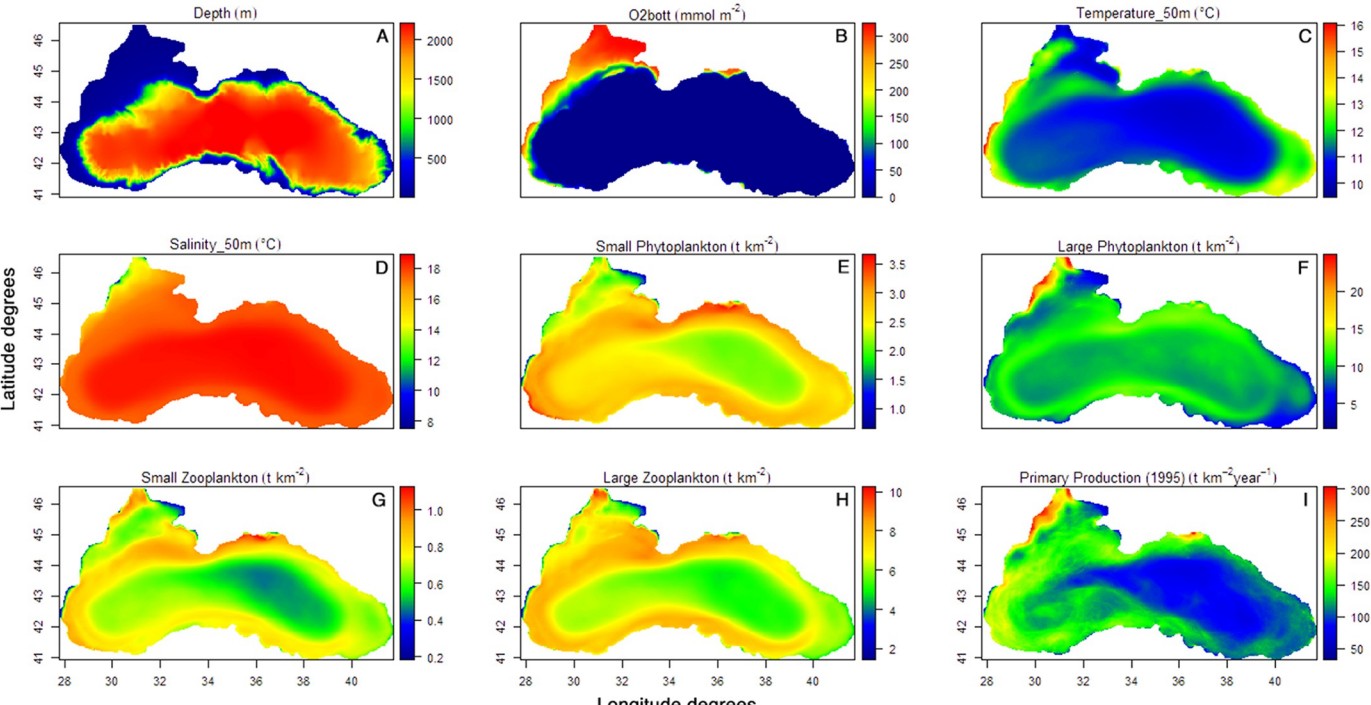

**Fig 1. Black Sea environmental and LTL biological outputs used as inputs and drivers in the HTL model.** (A) Depth (m); (B) Bottom oxygen (averaged over 1995–2021); (C) Temperature averaged over 50m (averaged over 1995–2021); (D) Salinity averaged over 50m (averaged over 1995–2021); (E) Small phytoplankton integrated over 50m (averaged over 1995–2021); (F) Large phytoplankton integrated over 50m (averaged over 1995–2021); (G) Small zooplankton integrated over 50m (averaged over 1995–2021); (H) Large zooplankton integrated over 50m (averaged over 1995–2021); (I) Primary production integrated over 50m.

validated against AquaMaps suitability maps, downloaded and standardised for the Black Sea domain [80] (https://www.aquamaps.org/).

## 3. Results and discussion

### 3.1. Ecopath

We modelled the entire Black Sea excluding the Azov Sea, covering approximately 423,000 km$^2$ with a maximum depth of 2,212 meters.

The final Blue2MF Black Sea HTL model, representing a snapshot of the ecosystem in 1995, has 47 functional groups (FGs), consisting of three dolphin species, three seabird groups, 24 fish groups, eight invertebrates groups, five planktonic groups, seagrass and seaweed, discards and detritus, and eight fishing fleet types (Fig 2, flow diagram). The model was balanced following the PREBAL ecological and thermodynamic rules (S2 Fig in S1 Appendix). The full list of species/FGs and the basic estimates of the balanced model are given in Table 1, while details on input data calculations and references per FGs are presented in S1 Table in S1 Appendix. Other inputs (landings, discards, effort and diet in S2 Appendix).

**3.1.1. Ecosystem indicators.** Functional groups are ranked by their TLs ranging from 1 (primary producers) to 4.29 (marine mammals); the highest TLs were found for 'Common dolphin' followed by 'Bottlenose dolphin', and 'harbour porpoise' (TL ≥ 4) (Table 1). 'Seagulls & cormorants' and 'terns', despite being considered top predators, showed relatively low TLs (3.38 and 3.36 respectively) due to the presence of discards and zooplankton in their diet. In

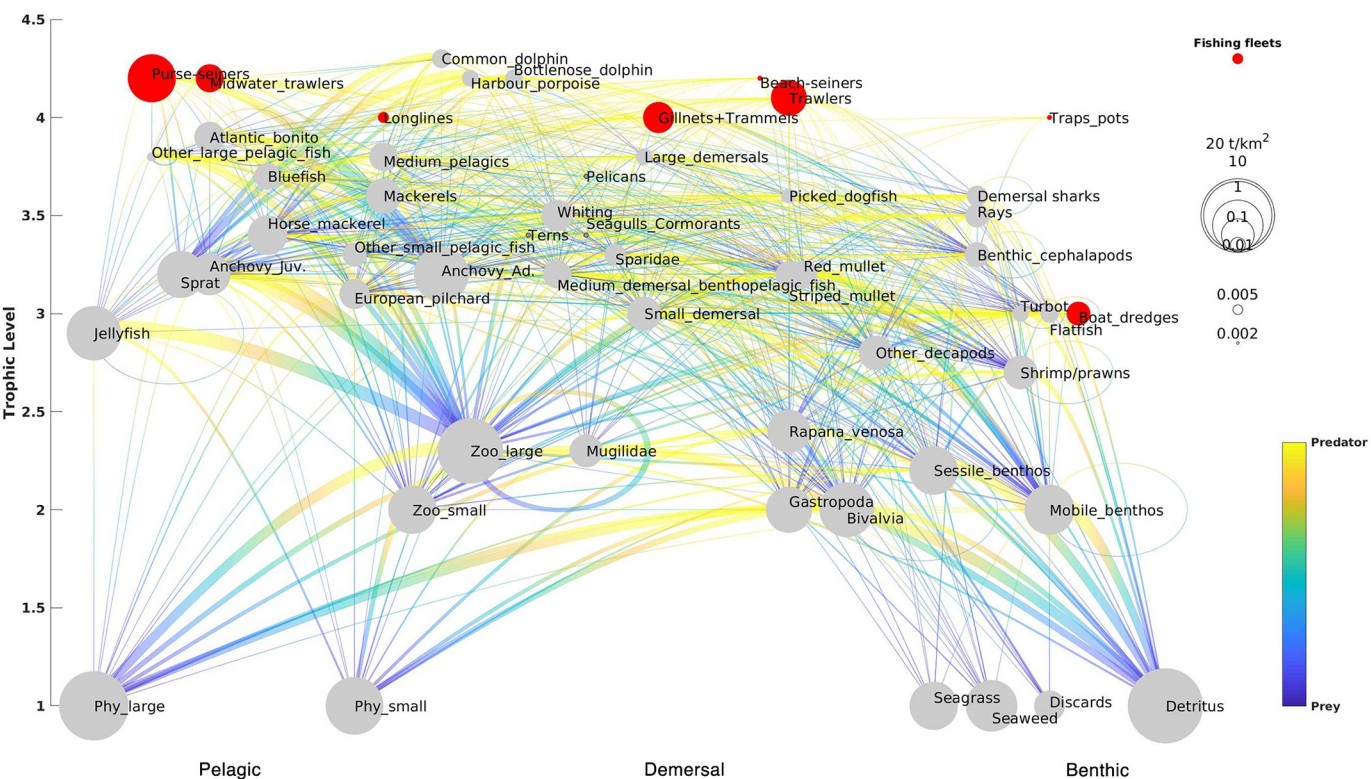

**Fig 2. Energy flow and biomass diagram of the Black Sea ecosystem (1995).** Functional groups (grey) and fleets (red) are represented by nodes. The relative sizes of the nodes for functional groups are proportional to their biomasses in the ecosystem, whereas the relative sizes of the nodes for fishing fleets are proportional to their catch volumes (fishing fleets: Beach-Seiner, Trawler, Dredge, Gillnet and Trammel net, Longline, Traps and pods, Purse Seine, Midwater Trawl). The functional groups are ordered by their trophic levels (y-axis) and their preferred domain (bottom x-axis Pelagic vs Demersal vs Benthic). Predator–prey relationships are expressed with a gradient yellow (predator) to blue (prey) line. The graph was created by the authors using MATLAB vR2014b.

EwE, discards are assigned to a detritus group with TL = 1 and seabirds feed considerably on discards, lowering the TL of those groups [81]. Several groups belonging to the pelagic and demersal domains had a TL greater than 3, while most of the functional groups with a TL smaller than 2 belonged to the benthic domain (Table 1 and Fig 2).

Other ecosystem indicators are listed in Table 2. Trophic level of the community (TL>1) was 2.5, which was slightly higher than that in the Mediterranean Sea in 2000 but lower than that in the Black Sea ecosystem in 1960. The TL of the catches in 1995, which indicates the exploitation status of fish populations [54], was much lower (2.97) than that in the Black Sea in the 1960s (3.52) and the 1980s (3.08), highlighting the continuous degradation of this ecosystem [16, 20].

The relative proportions of system throughput (TST) were in agreement with previous findings [16, 20], and were composed of consumption flows (38%) and respiration flows (23%), followed by flow to detritus (22%) and exports (17%). Comparing the absolute values of the total system throughput is not straightforward because this index is sensitive to the model structure [82]. Previous Black Sea models (Table 2) reported a TST of ~10,210 t km$^{-2}$ yr$^{-1}$ for 1995 and ~2007 t km$^{-2}$ yr$^{-1}$ for 1960; however, these models showed simpler structures than the model developed in this study, with only 10 and 22 functional groups, respectively. The TST value calculated in this study (Table 2) concurred with the value for the Mediterranean Sea model [81] that had the same model structure.

**Table 2. Summary statistics for the Black Sea and Mediterranean Sea.**

| Indicators | Black Sea 1995 (this study) | Black Sea 2000 (Akoglu et al, 2014) | Black Sea 1960 (Akoglu 2023) | Mediterranean Sea 2000 (Piroddi et al., 2015) |
|---|---|---|---|---|
| Mean TL_community (TL>1) | 2.50 | - | 2.77 | 2.18–2.36 |
| Mean TL_catches (TL>1) | 2.97 | 3 | 3.52 | 3.08 |
| TST (t km$^{-2}$ yr$^{-1}$) | 4008 | 10210 | 2007 | 4000 |
| TPP/TR | 1.77 | 1.16 | 1.1 | 5.55 |
| Mean TE (%) | 6.01 | 7.4 | 13.1 | 9.2 |
| SOI | 0.22 | 0.12 | 0.15 | 0.27 |
| Ascendency (%) | 27.50 | 31.7 | 26.5 | 42.9 |
| Overhead (%) | 72.50 | 68.3 | 73.5 | 57.1 |

TL = trophic level, TST = total system throughput, TPP/TR = total primary productivity to total respiration, TE = transfer efficiency, SOI = system omnivory index.

The ratio of total primary productivity to total respiration (TPP/TR), which is expected to be close to 1 in developed ecosystems, is an indication of ecosystem maturity [51]. Although the Black Sea was in a more mature state than the Mediterranean Sea (Table 2), our results showed that the Black Sea was losing this balance as TPP/TR increased since the 1960s. The transfer efficiency (TE) has also been decreasing over time in the Black Sea ecosystem, indicating inefficient functioning of the food web in terms of energy transfers to higher trophic levels. This could be a reason for the lack of recovery of higher-trophic-level predatory fish species in the ecosystem in addition to their overexploitation by fisheries.

The system omnivory index (SOI) indicates the range of feeding interactions in the food web; therefore, this index is also sensitive to the model structure, which is confirmed by our results showing a similar SOI with the Mediterranean Sea model [81].

Results of system ascendency and overhead, which represent the organizational degree in an ecosystem and its resilience against stress [55, 83], indicated higher values for both indices for the Black Sea than for the Mediterranean Sea (Table 2).

Heymans et al. [82] highlighted that a direct comparison of absolute indicator values generated by EwE across different ecosystems should take ecosystem traits such as type, size, depth, and location as well as model's topology into consideration. In this study, we compared these indicators with a Mediterranean model built with the same model structure [24] and with other Black Sea EwE models built with different structures for different periods. The overall analysis of these statistical indices revealed that TST and SOI, as well as indices based on the trophic level greater than 1 (either TL of the community and catches) and transfer efficiency, were comparable with similar model structures, while TPP/TR, ascendency, and overhead were more ecosystem-dependent. However, our comparisons indicated a more deteriorated ecosystem condition in the Black Sea with respect to the results provided by earlier modelling studies.

MTI analysis (Fig 3) revealed that the majority of predators exerted a direct negative impact on their prey due to diet preferences. Functional groups negatively impacted themselves due to cannibalism or inter-species competition. Planktonic groups had predominantly positive effects on all pelagic and demersal groups in the system (e.g., through a bottom-up effect). However, these planktonic groups had negative impacts on benthic groups, as they were preyed upon by the aforementioned groups (e.g., through a top-down effect).

The results also identified strong direct and indirect negative impacts of gillnets & trammels on dolphins (directly as by-caught by this fleet, and indirectly for targeting the same "prey"

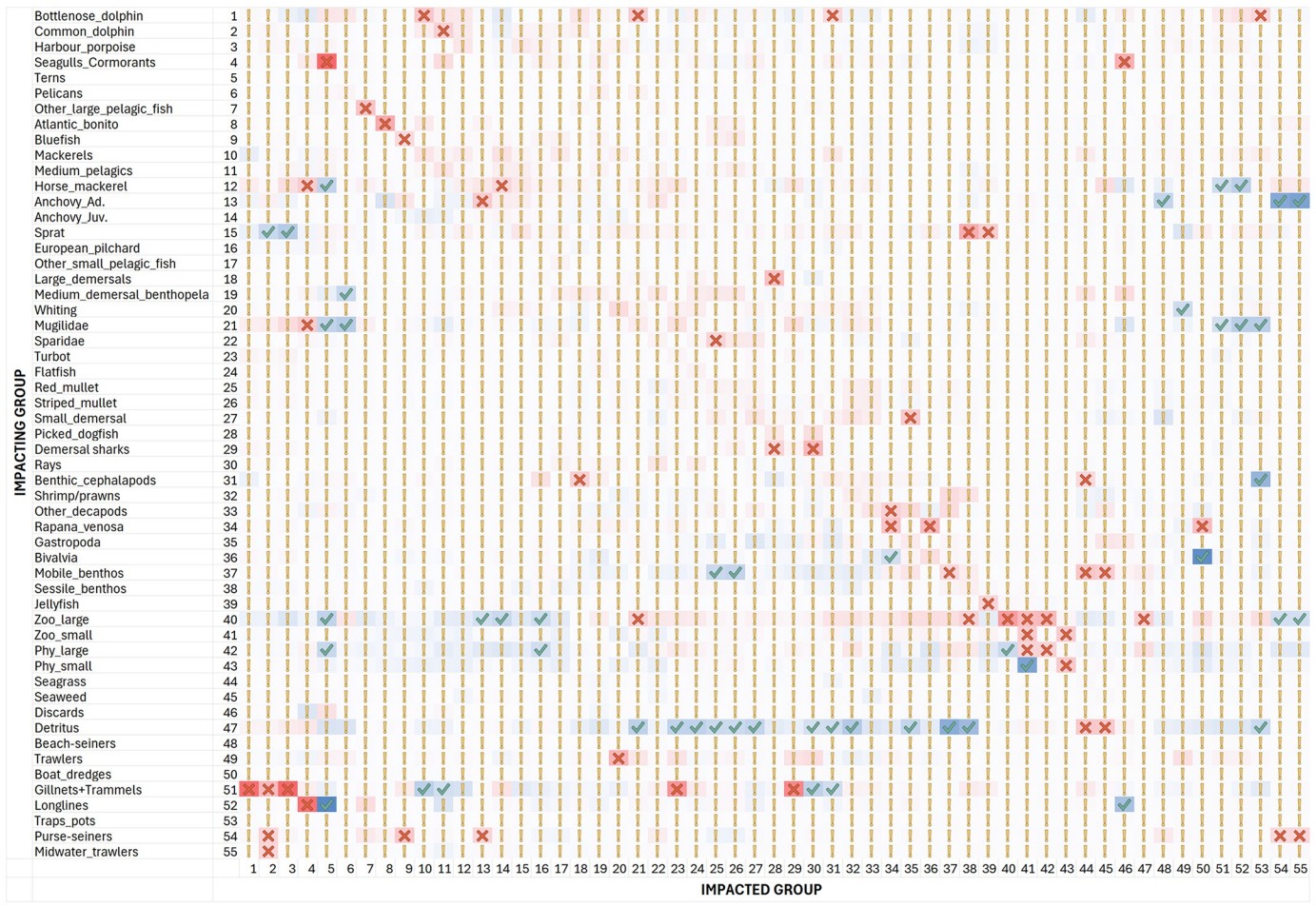

**Fig 3. Mixed trophic impact relationships between functional groups and fisheries.** Positive values (from white to blue) indicate positive impacts, and negative values (from white to red) indicate negative impacts. "Green check mark" symbol indicates MTI value greater than 0.2, "yellow exclamation mark" symbol indicates MTI value between 0.2 and -0.2, "red cross mark" symbol indicates MTI value smaller than -0.2.

species) and on turbot and demersal sharks. An indirect positive effect was found for prey species of the above groups. Trawlers showed a positive indirect impact on seagulls that feed on discards, and a negative direct impact on the main trawling target demersal species. Longlines had a negative impact on seagulls & cormorants as by-caught by this fleets (Appendix 2 -Discards), and an indirect positive impact on terns due to their competition with seagulls. Purse-seiners had direct negative impacts on bluefish and anchovy as well as on common dolphins by direct by-catches, but also indirectly, as this fleet targets dolphin's prey. It is interesting to note that, fishing fleets are substantially acting as top-predators exhibiting negative effects on their target species as "preys" and competing with other top-predators such as mammals and seabirds. Declining fish stocks determined an increase of interactions between top-predators and fishers, in particular, interactions between gillnets/trammels and dolphins are well documented in the Black Sea as well as in the Mediterranean Sea [84–86] having negative consequences for both cetaceans and fishers [87, 88]. As for the interaction between seabirds and fisheries, despite the lack of scientific evidence in the Black Sea region [89], several studies have documented this phenomenon worldwide. For example, negative interactions between longline fisheries and seabirds is well documented in the Atlantic and Pacific Oceans [90, 91]

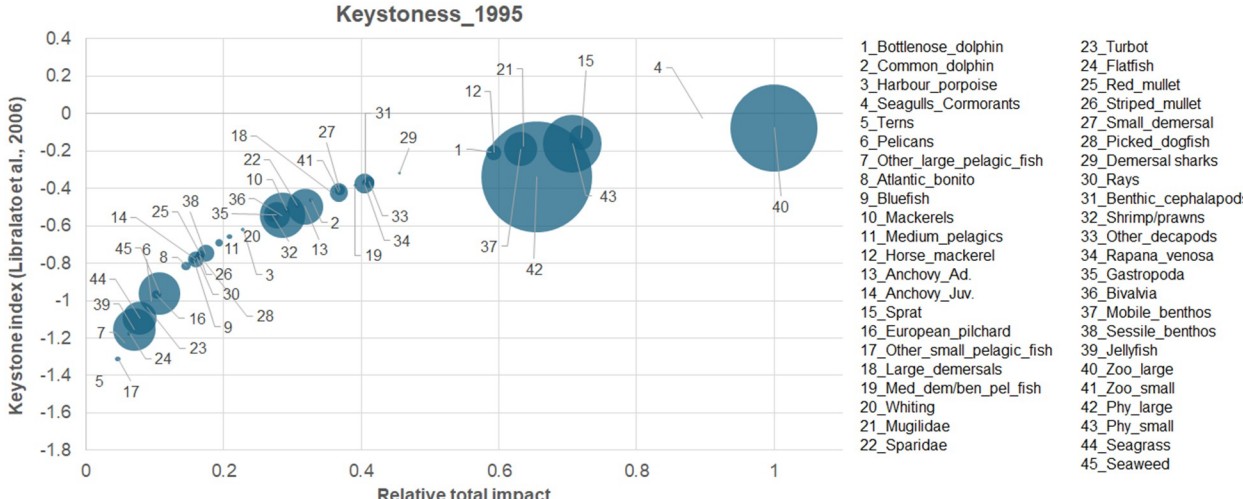

**Fig 4. Relative total impact ($\varepsilon_i$) versus keystoneness ($KS_i$).** The size of the circles is proportional to the species/group biomass.

whilst the positive effect of trawler's discards on gulls is also well reported in the North Sea [92, 93].

The species with the highest keystone index value (Fig 4) in the system was large zooplankton (40); however, by definition, keystone species are the species with a relatively low biomass, but their biomass changes would have a disproportionately large effect on the ecosystem structure [94]. According to this definition, seagulls& cormorants (4), followed by sprat (15), mugilidae (21), mobile benthos (37), horse mackerel (12) and bottlenose dolphin (1) were the functional groups with the highest keystones index values. In the Black Sea model, representing the early 1960s, demersal sharks and dolphins were the keystone species [16]. This contrasting finding indicated that a change of control on the food web occurred since the 1960s, from a top-down to a bottom-up control.

## 3.2. Ecosim

The fitting routine improved the model performance by 24% compared to the baseline fishing scenario (Table 3). Fig 5 and S3 Fig in S1 Appendix displayed the best fitted model predictions (lines) plotted against observations (points) when available for the 1995–2021 period, showing the fitting for stock assessed species (Fig 5) horse mackerel, anchovy, sprat, red mullet, whiting, turbot, picked dogfish and *Rapana venosa*.

**Table 3. Comparison across selected stepwise fitting interactions.**

| AIC test | Number of years | Number of observations | Number of estimated vulnerabilities (K) ($K_{max}$ = 38) | SS | AICc | Contribution to SS fitting |
|---|---|---|---|---|---|---|
| Fishing baseline | 27 | 1029 | 0 | 131.4 | -2117.7 | - |
| Best fitted "by-predator" | 27 | 1029 | 33 | 100.6 | -2322.7 | 23.5% |
| Hybrid fitting | 27 | 1029 | 5 | 99.7 | -2328.4 | 0.9% |

Model fishing baseline and the two fitting steps, showing the number of total parameters estimated (K, Vulnerabilities (Vs)), number of years of the time-series, number of observations, model sum of squares (SS), SS percentage of contribution to the fitting, and second-order Akaike Information Criterion (AIC$_c$).

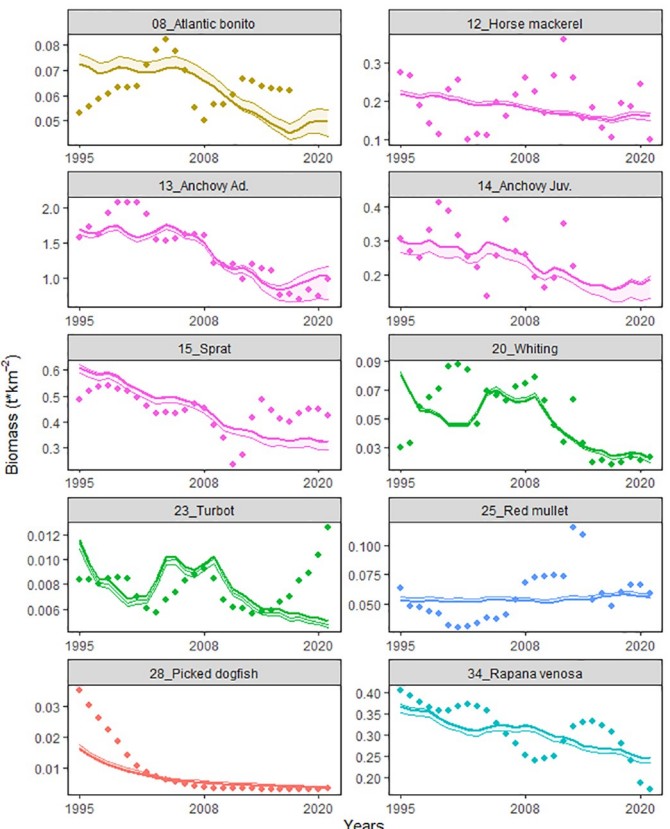

**Fig 5. Modelled biomass predictions of the best-fitted model.** Predictions (lines) versus observed (points) estimated by statistical stock assessments for the 1995–2021 period.

The fitted model showed an overall decreasing trends: all the commercially exploited species except red mullet and turbot declined since 1995 (Fig 5). Good agreements of the model fits with observed fluctuations were found for anchovy adults (SS = 0.89), sprat (SS = 1.08) and *Rapana venosa* (SS = 0.74), prediction was also good for Atlantic bonito (SS = 1.18) whilst a higher sum of squares was found for horse mackerel (SS = 4.13) due to the more scattered pattern of the observations. The model missed on representing the initial trends of the time series for whiting (SS = 3.65) and the steep observed decline of picked dogfish (SS = 6.04) whilst for turbot (SS = 2.10) the prediction did not replicate the end of the observed time-series. Red mullet (SS = 3.50) did not show clear trends neither for the model prediction and the observations.

According to the trend analysis, all the simulated biomasses except juvenile anchovy had significant decreasing trends (Fig 5 and S4 Table in S1 Appendix). The overall declines could be attributed to poorly-managed fisheries at national level and the lack of a basin-wide management policy. The Black Sea fish stocks are in fact shared across countries due to the high level of connectivity and the migratory behaviours of the primarily exploited species [95]. Stock assessment for picked dogfish (observations in Fig 5) showed a very steep declining trend since the 90's, due to over-exploitation and the use inappropriate fishing gear [96]: the observed and modelled trend agrees with the classification of "vulnerable" (high risk of extinction in the wild) given by the IUCN Red List Threatened Species for *Squalus acanthias* in the Black Sea region [97].

Red mullet was the only stocked assessed species that did not show a clear predicted declining trend, with highly scattered observed calibration data (Fig 5). Similarities between congeneric species *Mullus barbatus* and *Mullus surmuletus* (red mullet and striped mullet respectively) often lead to 'voluntary' inaccuracies in species reporting by fishermen [98, 99]. Red mullet are often reported as striped mullet that have a smaller minimum sampling size to avoid penalties and to land more catch. This practice raises concerns about the accuracy of catch statistics for these two species.

The model was unable to reproduce the recent increase in turbot biomass (2017–2021) predicted by the stock assessment. This increase could be captured by forcing the observed turbot catches for those years (S4 Fig in S1 Appendix). This model uses the observed total catches per functional group for all countries fishing in the Black Sea (Ukraine, Romania, Bulgaria, Russia, and Turkey). Overfishing has caused a strong decline in turbot since 1980; for this reason, since 2016, a fishing management plan has been put in place by the European Commission for Romania and Bulgaria in the Black Sea [100]. Turkey, the main fishery country for turbot in the Black Sea, did not support these management restrictions. Accordingly to Hulak et al (2021), the decrease in turbot landings reported in the Turkish catch statistics, after 2002, is a consequence of the effort of riparian countries (Ukraine, Romania, and Bulgaria) to stop illegal Turkish turbot fisheries in their EEZ. We presume that abovementioned restrictions/controls could be the consequence of the increase in biomass predicted by stock assessment models.

Ongoing efforts have been made to manage and conserve key fish stocks by European Commission who aiming to achieve sustainable fishery for the Mediterranean and Black Seas in line with the 2030 Strategy of the GFCM. Recent management measures proposals includes total allowable catches (TAC) and quotas for turbot and catch limits also for sprat [101].

## 3.3. Ecospace

Ecospace output represented the species distribution of the functional groups driven by prey distributions, and avoidance of predators (including fisheries) within their preferred habitat preference (e.g., depth, oxygen) (S3 Table in S1 Appendix). It should be noted that, the output distributions of the modelled functional groups should be considered with caution, due to the lack of spatial data for many Black Sea species. Yet, qualitative validation can be cross-checked with available information. For instance, the predicted spatial biomass distributions of marine mammals (S5 Fig in S1 Appendix, upper panel) in shallower regions for bottlenose dolphins and harbour porpoises and offshore areas for common dolphins are in line with aerial surveys and species distribution models [102, 103]. Seabirds showed distributions over waters shallower than 500 m (S5 Fig in S1 Appendix, lower panel), mainly in the North and West of the basin where their preys are distributed following a similar pattern as observed by Piroddi et al. [26] for the neighbouring Mediterranean Sea.

The distributions of commercial migratory pelagic species, such as Atlantic bonito, bluefish and the mackerels groups were only driven by their prey distribution as migration patterns of these species were not considered in this study (Fig 6 and S6 Fig in S1 Appendix).

Among the stock-assessed species (Fig 6), sprat and anchovy were the species that mainly drove the distribution of many pelagic predators in the model.

The anchovies exhibit a strong seasonal pattern in which, in winter, they migrate between the North-Western shelf and the south of the basin. Notably, over winter, the highest anchovy biomass and a significant portion of anchovy catches are associated with the southeast region [95, 104]. Annual averaged maps presented here may not represent these spatial seasonal patterns successfully; however, overall, our predicted distributions agreed well with those produced by stock assessment models [95, 104].

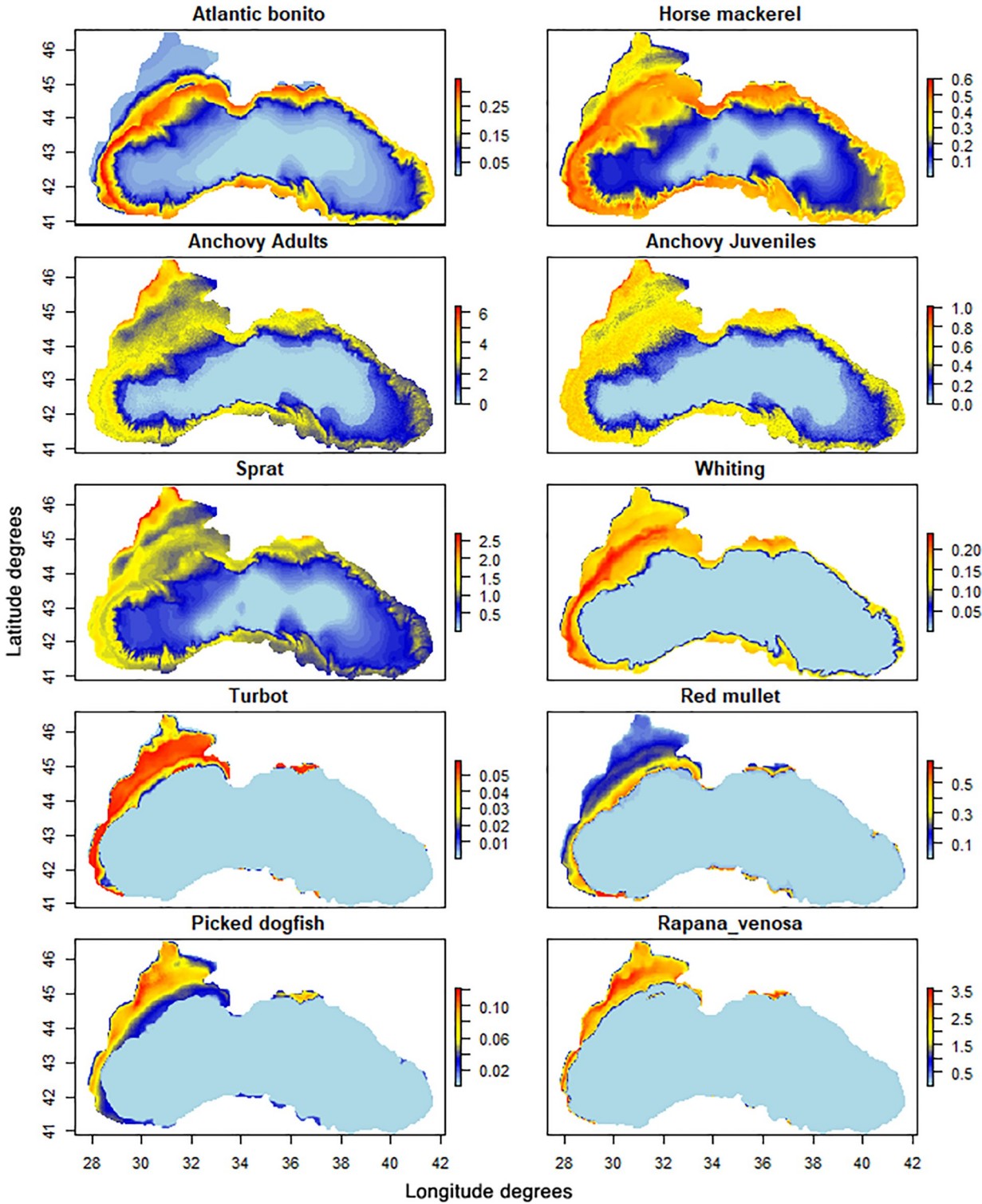

**Fig 6. Biomass spatial distributions of the stock-assessed species.** Data are presented as t*km$^{-2}$ averaged over all modelled years 1995–2021.

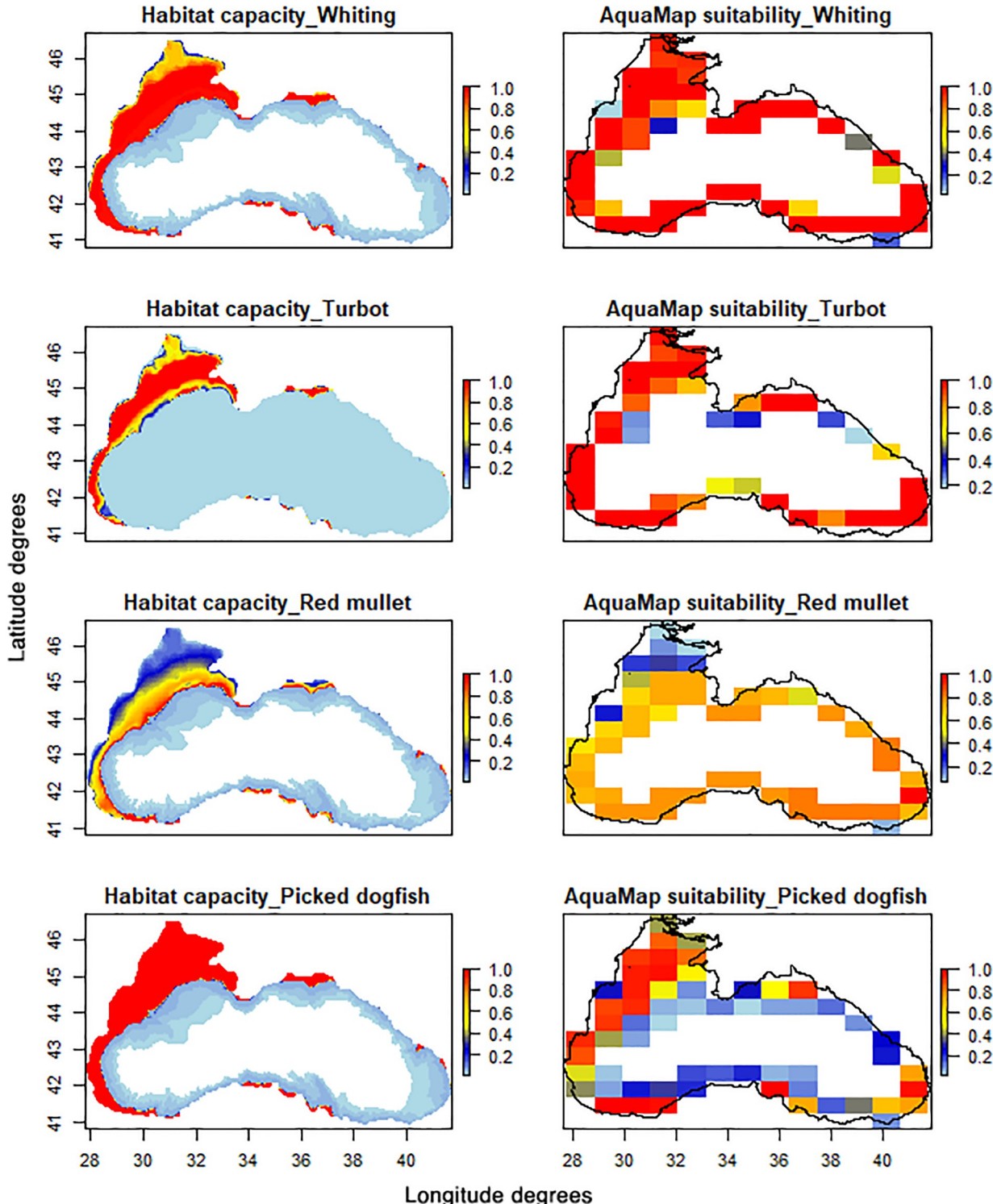

**Fig 7. Qualitative spatial validation of stocked assessed species.** Habitat capacity computed by Ecospace model (left panels) versus AquaMaps suitability maps (right panels).

Benthic species had a confined distribution (Fig 6 and S6 Fig in S1 Appendix) over the continental shelf, due to the anoxic conditions that occur in deeper waters where the annual bottom oxygen concentration is lower than 10 mmol $O_2/m^2$ (Fig 1C), whilst many demersal groups have confined distributions for trophic relationships on benthos. The spatial distribution of picked dogfish and turbot (Fig 6) well agreed with previously reported data [96, 105].

A qualitative validation of the Ecospace habitat foraging capacities (HFCs) of stock assessed demersal species whiting, turbot, red mullet and picked dogfish showed an overall agreement with the maps of habitat suitability model extracted from AquaMaps [80] (Fig 7). Differences such as lower habitat capacity values towards the centre of the basin could be due to the different model resolutions between Ecospace and AquaMaps and an overestimation of the species sensitivity at lower bottom oxygen concentration.

Overall, our results confirmed the highest biodiversity confined in the North-Western shelf, particularly in the shallow neritic waters [22], highlighting even further the necessity to prioritize spatial management actions to preserve the remaining biodiversity.

## 4. Conclusions

Since the 1960s, the Black Sea has undergone a series of transformations driven by numerous anthropogenic stressors. The current military aggression happening in Ukraine, which impacts coastal and marine ecosystems of the region, for example via chemical/noise pollution, habitats damage (from shelling and fortifications), and limitation of conservation activities [106], necessitates even more the need of a temporal and spatial assessment tool enable to evaluate the single and cumulative impacts of these pressures.

This new model and its findings constitute the reference baseline (defined as the past and current status) for the Black Sea ecosystem. The model well represent the overall temporal declining trends of the main fishery target species and offers the first attempt on representing species spatial patterns within an ecosystem-food-web framework. The model will be used for testing policy management scenarios set out by EU legislation, for example, applying fishing measures (i.e., total allowable catch [TAC] for turbot/sprat) coming from the Common Fisheries Policy and the GFCM; and restoration/conservation measures (i.e., MPAs, eutrophication reduction) following the recent Nature Restoration Law, the Biodiversity Strategy, and the MSFD.

Such efforts are needed to directly assist spatial management actions (e.g., by prioritizing specific areas of concern) and support management decisions for current and future European policies and international agreements/legislations.

## Supporting information

**S1 Appendix.**
(DOCX)

**S2 Appendix. Model inputs.** Landings, Discards, Relative effort, Diet.
(XLSX)

## Acknowledgments

We thank Dr Andrea Pierucci from the European Commission, Joint Research Centre in Ispra for the support and guidance on stock assessment data.

## Author Contributions

**Writing – review & editing:** Natalia Serpetti, Chiara Piroddi, Ekin Akoglu, Elisa Garcia-Gorriz, Svetla Miladinova, Diego Macias.

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
