## [Decision Letter · Decision Letter 0]

13 Aug 2024

PONE-D-24-26903The state of the art modelling for the Black Sea ecosystem to support European policiesPLOS ONE

Dear Dr. Serpetti,

Thank you for submitting your manuscript to PLOS ONE. After careful consideration, we feel that it has merit but does not fully meet PLOS ONE’s publication criteria as it currently stands. Therefore, we invite you to submit a revised version of the manuscript that addresses the, mostly minor, points raised during the review process. Please ensure that your decision is justified on PLOS ONE’s publication criteria and not, for example, on novelty or perceived impact.

We look forward to receiving your revised manuscript.

Kind regards,

Athanassios C. Tsikliras

Academic Editor

PLOS ONE

Journal Requirements:

Reviewers' comments:

Reviewer's Responses to Questions

**Comments to the Author**

1. Is the manuscript technically sound, and do the data support the conclusions?

Reviewer #1: Yes

Reviewer #2: Yes

2. Has the statistical analysis been performed appropriately and rigorously? 

Reviewer #1: Yes

Reviewer #2: Yes

3. Have the authors made all data underlying the findings in their manuscript fully available?

Reviewer #1: Yes

Reviewer #2: Yes

4. Is the manuscript presented in an intelligible fashion and written in standard English?

Reviewer #1: Yes

Reviewer #2: Yes

5. Review Comments to the Author

Reviewer #1: Thank you for the opportunity to review this insightful manuscript titled "The state of the art modelling for the Black Sea ecosystem to support European policies." The study presents a significant contribution to our understanding of the Black Sea ecosystem, particularly in the context of supporting European policy frameworks. The methodology is rigorous, and the findings are both relevant and timely.

While the manuscript is well-structured and the content is compelling, I have identified a few minor issues that could be addressed to further enhance the clarity and impact of the paper. These minor suggestions are detailed below.

The species mentioned in the introduction should be supported with their scientific names, as was done with Mnemiopsis leidyi. While I understand that these are provided in the appendices, not all readers may have the same access to the full text or appendices, leading to potential misunderstandings.

In most of the sentences referenced in the introduction, a single reference is used, which weakens the strength of the literature support.

Line 44-45: The authors mention 'recent reports,' implying multiple sources; however, only a single reference is provided.

I would not describe the Black Sea as a 'hotspot' of biodiversity, as stated in lines 56-57. While there is certainly biodiversity present in the Black Sea, I do not agree that it qualifies as a hotspot compared to other seas.

In lines 60 to 76, in addition to mentioning the contributions of the European Commission and its science Directorate-General (DG) and the Joint Research Centre, it would be beneficial to also highlight the efforts of the GFCM in bringing countries to the table and in data collection. These efforts are particularly relevant, given that the publication utilizes significant data from these initiatives.

Lines 435-443 should be supported by existing literature. It is important to verify whether the 'assumptions' familiar in other seas also hold true for the Black Sea. For example, it is well-documented that interactions between dolphins and gillnets & trammels are frequent in the Black Sea due to declining fish stocks, which increases encounters between fishermen and dolphins. However, I have not come across any publications regarding the impact of trawlers on seagulls in the Black Sea. If such references exist, they should be cited here.

In lines 491-503, it would be important to mention the quota system recommended and implemented by the GFCM to protect the remaining turbot stocks. Member countries of the GFCM are adhering to this regulation. The relevant text is as follows:

In lines 481-490, it should be noted that the similarities between Mullus barbatus (red mullet) and Mullus surmuletus (striped red mullet) often lead to inaccuracies in species reporting by fishermen. For instance, in Turkey, the minimum landing size for Mullus barbatus is 13 cm, while for Mullus surmuletus it is 11 cm. To avoid penalties and to land more catch, fishermen may register Mullus barbatus as Mullus surmuletus. This practice raises concerns about the accuracy of catch statistics for these two species.

'In the Black Sea, the proposal includes catch limits and quotas for sprat and turbot. For sprat, the Commission proposes to maintain the 2023 catch limit. For turbot, the levels of total allowable catches (TAC) and quotas will be set and adopted at the 2023 GFCM annual session. The proposal demonstrates the Commission's objective to make fisheries in these two sea basins sustainable, in line with the 2030 Strategy of the General Fisheries Commission for the Mediterranean (GFCM), which is the regional fisheries management organisation competent for the conservation and management of fish stocks in the Mediterranean and the Black Seas.' (https://ec.europa.eu/commission/presscorner/detail/en/ip_23_4861)

This addition would emphasize the ongoing efforts to manage and conserve key fish stocks in the Black Sea.

The manuscript's contribution to supporting European policies should be elaborated further in the discussion section, particularly given its mention in the title. Expanding on how the findings align with or inform key European policy objectives would strengthen the manuscript's relevance. Specifically, discussing how the research supports the 2030 Strategy of the General Fisheries Commission for the Mediterranean (GFCM) and other EU sustainability goals would provide valuable context. This could include an analysis of how the data contributes to decision-making processes at the European level. Such a discussion would emphasize the manuscript’s practical implications for policy development and implementation.

Reviewer #2: This study builds on previous EwE models of the Black Sea ecosystem, adding more refined data where possible, and develops the first spatially and temporally explicit Ecospace model for the region. Spatial results are qualitatively compared to other published and modeled species distributions for verification purposes in the absence of detailed spatial data for many Black Sea species. The developed Ecospace is expected to be the baseline for future testing of spatial management scenarios in support of and potential revision of the MSFD. I find the study to be rigorous and well documented, with a detailed supplement. I suggest that the authors do some mostly minor revisions considering the comments I have made on the attached PDF.

6. PLOS authors have the option to publish the peer review history of their article (what does this mean?). If published, this will include your full peer review and any attached files.

Reviewer #1: **Yes: **Taner Yildiz

Reviewer #2: No

---

## [Author Response · Author response to Decision Letter 0]

13 Sep 2024

We sincerely thank the reviewers for their detailed feedback on our paper, which has greatly enhanced its quality. We found the reviewers’ comments to be incredibly helpful, particularly regarding the technical aspects of our modelling approach and insights into the Black Sea ecosystem.

We have carefully followed all the suggestions and have made the necessary revisions to the manuscript accordingly. 

Please refer to "Response to Reviewers" file for details.

Best Regards,

Natalia Serpetti and co-Authors.

---

## [Decision Letter · Decision Letter 1]

2 Oct 2024

State of the art modelling for the Black Sea ecosystem to support European policies

PONE-D-24-26903R1

Dear Dr. Serpetti,

We’re pleased to inform you that your manuscript has been judged scientifically suitable for publication and will be formally accepted for publication once it meets all outstanding technical requirements.

Kind regards,

Athanassios C. Tsikliras

Academic Editor

PLOS ONE

Additional Editor Comments (optional):

Reviewers' comments:

Reviewer's Responses to Questions

**Comments to the Author**

1. If the authors have adequately addressed your comments raised in a previous round of review and you feel that this manuscript is now acceptable for publication, you may indicate that here to bypass the “Comments to the Author” section, enter your conflict of interest statement in the “Confidential to Editor” section, and submit your "Accept" recommendation.

Reviewer #2: All comments have been addressed

2. Is the manuscript technically sound, and do the data support the conclusions?

Reviewer #2: Yes

3. Has the statistical analysis been performed appropriately and rigorously? 

Reviewer #2: Yes

4. Have the authors made all data underlying the findings in their manuscript fully available?

Reviewer #2: Yes

5. Is the manuscript presented in an intelligible fashion and written in standard English?

Reviewer #2: Yes

6. Review Comments to the Author

Reviewer #2: Good job on addressing reviewer comments or responding to them accordingly. I spotted several grammatical inconsistencies in the added text, so I suggest a careful language check of these parts or even better the whole manuscript before publication, which will strengthen the paper.

7. PLOS authors have the option to publish the peer review history of their article (what does this mean?). If published, this will include your full peer review and any attached files.

Reviewer #2: No

---

## [Editor Report · Acceptance letter]

16 Oct 2024

PONE-D-24-26903R1 

PLOS ONE

Dear Dr. Serpetti, 

I'm pleased to inform you that your manuscript has been deemed suitable for publication in PLOS ONE. Congratulations! Your manuscript is now being handed over to our production team.

Kind regards, 

on behalf of

Professor Athanassios C. Tsikliras 

Academic Editor

PLOS ONE